# Sparse sampling and tensor network representation of two-particle Green's functions

**Hiroshi Shinaoka[1]⋆, Dominique Geffroy[2,3], Markus Wallerberger[4,3], Junya Otsuki[5], Kazuyoshi Yoshimi[6], Emanuel Gull[4] and Jan Kuneš[3,7]**

**1** Department of Physics, Saitama University, Saitama 338-8570, Japan
**2** Department of Condensed Matter Physics, Faculty of Science,
Masaryk University, Kotlářská 2, 611 37 Brno, Czech Republic
**3** Institute of Solid State Physics, TU Wien, 1040 Vienna, Austria
**4** University of Michigan, Ann Arbor, Michigan 48109, USA
**5** Research Institute for Interdisciplinary Science,
Okayama University, Okayama 700-8530, Japan
**6** Institute for Solid State Physics, University of Tokyo, Chiba 277-8581, Japan
**7** Institute of Physics, Czech Academy of Sciences,
Na Slovance 2, 182 21 Praha 8, Czech Republic

⋆ shinaoka@mail.saitama-u.ac.jp

## Abstract

Many-body calculations at the two-particle level require a compact representation of two-particle Green's functions. In this paper, we introduce a sparse sampling scheme in the Matsubara frequency domain as well as a tensor network representation for two-particle Green's functions. The sparse sampling is based on the intermediate representation basis and allows an accurate extraction of the generalized susceptibility from a reduced set of Matsubara frequencies. The tensor network representation provides a system independent way to compress the information carried by two-particle Green's functions. We demonstrate efficiency of the present scheme for calculations of static and dynamic susceptibilities in single- and two-band Hubbard models in the framework of dynamical mean-field theory.

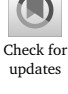

# 1 Introduction

Two-particle (2P) Green's functions are building blocks of a variety of many-body theories [1]. They are a key element for calculation of susceptibilities in the framework of dynamical mean-field theory (DMFT) [2] as well as for diagrammatic calculations including vertex corrections and diagrammatic extensions of DMFT [3–16]. In equilibrium finite-temperature formalism, the 2P quantities depend on three Matsubara frequencies, the low- and high-frequency parts of which must be treated accurately. Storing 2P quantities alone is a challenge, more so when multiple orbitals, low symmetries or low temperatures are involved.

Several methods have been proposed to address the storage issue. Conventional approaches are based on a separate treatment of the low- and high-frequency parts [17–21]. The frequency dependence is treated exactly in a small low-frequency box while in the outside region an asymptotic form is used. This works efficiently at relatively high temperatures. As the temperature is lowered the size of the low-frequency box grows, until it becomes prohibitively large.

Recently, the intermediate representation (IR) basis was introduced as a promising solution to the storage issue [22]. In IR the size of the data grows only logarithmically with the inverse temperature $\beta$ and the bandwidth. A fitting procedure allows IR expansion of numerical data in the Matsubara frequency domain. Nevertheless, two obstacles remain: the computational cost of the IR expansion and the size of the IR tensor.

Regarding the first obstacle, the input data for the fitting scheme is very large, having a dense support in the Matsubara frequency domain. The fitting procedure thus becomes prohibitively expensive at low $T$. As for the second one, the IR represents a 2P quantity as a high-order tensor involving spin and orbital dimensions, in addition to those for the IR basis itself. Further compactification of the tensor is required for solving realistic multi-orbital

systems at low $T$.

In this paper, we address these two issues. First, we introduce a sparse grid in the Matsubara frequency domain, which contains the desired information about the 2P Green's functions This extends the approach developed in Ref. [23] for one-particle (1P) Green's functions. We introduce an efficient tensor network representation of the IR tensor and a fitting (regression) algorithm to determine it. Reduction of the data to be sampled thanks to the sparse grid makes evaluation of the IR coefficients very efficient and solves the first issue. The tensor regression provides a model-independent way to compress the IR tensor and tackles the second issue.

We demonstrate the performance of the present method in the context of DMFT. First, we test the accuracy of sparse sampling and tensor network representation by calculating the static susceptibility of the single-band Hubbard model on a square lattice. Next, we show the efficiency of the present method for dynamical susceptibility calculation for a two-band Hubbard model with low symmetry.

The paper is organized as follows. In the next section, the IR for 1P and 2P Green's functions is reviewed. Sparse sampling of 2P Green's functions is introduced in Sec. 3. In Sec. 4, the tensor network representation is presented. Its accuracy for computing of static susceptibilities of the single-band Hubbard model is demonstrated in Sec. 5. In Sec. 6, we present numerical results for dynamical susceptibility calculations, in the more demanding context of an ordered phase of the two-band Hubbard model. In Sec. 7, we summarize and conclude.

## 2   Intermediate Representation (IR) for Green's functions

Here we review the IR for 1P and 2P Green's functions introduced in [22] and [24]. The reader may refer to Section 7 of [25] for a review.

### 2.1   One-particle Green's function

The IR for 1P Green's functions was introduced in Ref. [24]. The spectral (Lehmann) representation of the 1P Green's function $G(\tau)$ in the imaginary-time domain reads

$$G(\tau) = -\int_{-\omega_{\max}}^{\omega_{\max}} d\omega K^{\alpha}(\tau, \omega)\rho(\omega), \tag{1}$$

where we assume $\hbar = 1$. The superscript $\alpha$ specifies statistics: $\alpha = \text{F}$ for fermion and $\alpha = \text{B}$ for boson. The spectrum $\rho(\omega)$ is assumed to be bounded within the interval $[-\omega_{\max}, \omega_{\max}]$. The kernel $K^{\alpha}(\tau, \omega)$ reads

$$K^{\alpha}(\tau, \omega) \equiv \omega^{\delta_{\alpha,\text{B}}} \frac{e^{-\tau\omega}}{1 \pm e^{-\beta\omega}} \tag{2}$$

for $0 \leq \tau \leq \beta$ ($\beta$ is the inverse temperature). Here, the $+$ and $-$ signs are used for fermions and bosons, respectively. The extra $\omega$ factor for bosons in Eq. (2) is introduced in order to avoid a singularity of the kernel at $\omega = 0$.

For fixed values of $\omega_{\max}$ and $\beta$, the IR basis functions are defined through the singular value decomposition (SVD)

$$K^{\alpha}(\tau, \omega) = \sum_{l=0}^{\infty} S_l^{\alpha} U_l^{\alpha}(\tau) V_l^{\alpha}(\omega), \tag{3}$$

$$\text{with} \int_0^{\beta} d\tau U_l^{\alpha}(\tau) U_{l'}^{\alpha}(\tau) = \int_{-\omega_{\max}}^{\omega_{\max}} d\omega V_l^{\alpha}(\omega) V_{l'}^{\alpha}(\omega) = \delta_{ll'},$$

where the singular values $S_l^\alpha \, (> 0)$ decrease with increasing $l$ exponentially.

For fermions, the Green's function can be expanded as

$$G(\tau) = \sum_{l=0}^{\infty} \mathcal{G}(l) U_l^{\mathrm{F}}(\tau), \tag{4}$$

$$\mathcal{G}(l) = -S_l^{\mathrm{F}} \rho_l, \tag{5}$$

where $\rho_l \equiv \int_{-\omega_{\mathrm{max}}}^{\omega_{\mathrm{max}}} d\omega \rho(\omega) V_l^{\mathrm{F}}(\omega)$. The exponential decay of $S_l^{\mathrm{F}}$ ensures a fast decay of $G_l$ if the spectrum is bounded in $[-\omega_{\mathrm{max}}, \omega_{\mathrm{max}}]$. The accuracy of the expansion can be controlled by applying a cut-off for the singular values. The Matsubara-frequency representation of the Green's function reads

$$G(i\omega_n) = \int_0^\beta d\tau G(\tau) e^{i\omega_n \tau} = \sum_{l=0}^{\infty} \mathcal{G}(l) U_l^{\mathrm{F}}(i\omega_n) \tag{6}$$

$$\text{with } U_l^{\mathrm{F}}(i\omega_n) \equiv \int_0^\beta d\tau U_l^{\mathrm{F}}(\tau) e^{i\omega_n \tau}.$$

## 2.2 General form of IR for Two-particle Green's functions

The problem of the three- and four-point Green's functions was considered in Ref. [22]. The four-point Green's function can be expressed in the Matsubara domain as

$$G_{ijkl}(i\omega_n, i\omega_{n'}, i\nu_m)$$

$$\equiv \beta^{-2} \int_0^\beta d\tau_1 d\tau_2 d\tau_3 d\tau_4 e^{i(\omega_n \tau_{12} + \omega_{n'} \tau_{34} + \nu_m \tau_{14})} \langle T_\tau c_i^\dagger(\tau_1) c_j(\tau_2) c_k^\dagger(\tau_3) c_l(\tau_4) \rangle \tag{7}$$

$$\equiv \sum_{r=1}^{16} \sum_{l_1 l_2 l_3} \mathcal{G}_O(r, l_1, l_2, l_3) U_{l_1}^\alpha(i\omega) U_{l_2}^{\alpha'}(i\omega') U_{l_3}^{\alpha''}(i\omega''), \tag{8}$$

where $\omega$, $\omega'$, $\omega''$ stand for $r$-dependent combinations of $\omega_n$, $\omega_{n'}$, and $\nu_m$ listed in Table 1. Similarly, the indices $\alpha$, $\alpha'$ and $\alpha''$ take the value $F$ or $\bar{B}$ depending on $r$ as indicated in Table 1. The indices $i, j, k, l$ denote the flavor (combined spin and orbital), while $O$ is a composite index representing the quadruplet $(i, j, k, l)$. Formally, the tensor $\mathcal{G}$ contains full information about $G$ (meaning in particular, information about its values at all bosonic and fermionic frequencies for all flavors).

## 2.3 Simplified form for fixed bosonic frequency

We now derive a variant of Eq. (8), which holds for a fixed bosonic frequency. The principle of the derivation is the same as above. Nevertheless, if one is interested in only a few bosonic frequencies, the reduced number of degrees of freedom allows us to represent the 2P Green's function as a tensor of lower rank. In the conventional particle-hole notation, the Green's function depends only on two fermionic frequencies for a fixed bosonic frequency. It is shown in Appendix A that in this case, this frequency dependence can be written as

$$G_O(i\omega_n, i\omega_{n'}, i\nu_m) = \sum_{r=1}^{12} \sum_{l_1, l_2} \mathcal{G}_O(r, l_1, l_2; i\nu_m) U_{l_1}^{(r)}(i\omega) U_{l_2}^{(r)}(i\omega'), \tag{9}$$

where $\omega_n$ and $\omega_{n'}$ are fermionic frequencies, and $\nu_m$ is a bosonic frequency. The index $r$ relates to the 12 distinct representations generated from the three terms in Eq. (18). The

Table 1: 16 different notations of Matsubara frequencies and statistics for the four-point Green's functions $G$. Adapted from Ref. [22].

| $r$ | $(i\omega, i\omega', i\omega'')$ | $(\alpha, \alpha', \alpha'')$ |
|---|---|---|
| 1 | $(i\nu_m + i\omega_n, -i\omega_n, i\omega_{n'})$ | (F,F,F) |
| 2 | $(i\nu_m + i\omega_n, -i\omega_n, -i\nu_m - i\omega_{n'})$ | (F,F,F) |
| 3 | $(i\nu_m + i\omega_n, i\omega_{n'}, -i\nu_m - i\omega_{n'})$ | (F,F,F) |
| 4 | $(-i\omega_n, i\omega_{n'}, -i\nu_m - i\omega_{n'})$ | (F,$\bar{\text{B}}$,F) |
| 5 | $(i\nu_m + i\omega_n, i\nu_m, i\nu_m + i\omega_{n'})$ | (F,$\bar{\text{B}}$,F) |
| 6 | $(i\nu_m + i\omega_n, i\nu_m, -i\omega_{n'})$ | (F,$\bar{\text{B}}$,F) |
| 7 | $(i\nu_m + i\omega_n, i\nu_m + i\omega_n + i\omega_{n'}, i\nu_m + i\omega_{n'})$ | (F,$\bar{\text{B}}$,F) |
| 8 | $(i\nu_m + i\omega_n, i\nu_m + i\omega_n + i\omega_{n'}, i\omega_n)$ | (F,$\bar{\text{B}}$,F) |
| 9 | $(i\nu_m + i\omega_n, i\omega_n - i\omega_{n'}, -i\omega_{n'})$ | (F,$\bar{\text{B}}$,F) |
| 10 | $(i\nu_m + i\omega_n, i\omega_n - i\omega_{n'}, i\omega_n)$ | (F,$\bar{\text{B}}$,F) |
| 11 | $(-i\omega_n, i\nu_m, i\nu_m + i\omega_{n'})$ | (F,$\bar{\text{B}}$,F) |
| 12 | $(-i\omega_n, i\nu_m, -i\omega_{n'})$ | (F,$\bar{\text{B}}$,F) |
| 13 | $(-i\omega_n, -i\omega_n + i\omega_{n'}, i\nu_m + i\omega_{n'})$ | (F,$\bar{\text{B}}$,F) |
| 14 | $(-i\omega_n, -i\nu_m - i\omega_n - i\omega_{n'}, -i\omega_{n'})$ | (F,$\bar{\text{B}}$,F) |
| 15 | $(i\omega_{n'}, i\nu_m + i\omega_n + i\omega_{n'}, i\nu_m + i\omega_{n'})$ | (F,$\bar{\text{B}}$,F) |
| 16 | $(i\omega_{n'}, -i\omega_n + i\omega_{n'}, i\nu_m + i\omega_{n'})$ | (F,$\bar{\text{B}}$,F) |

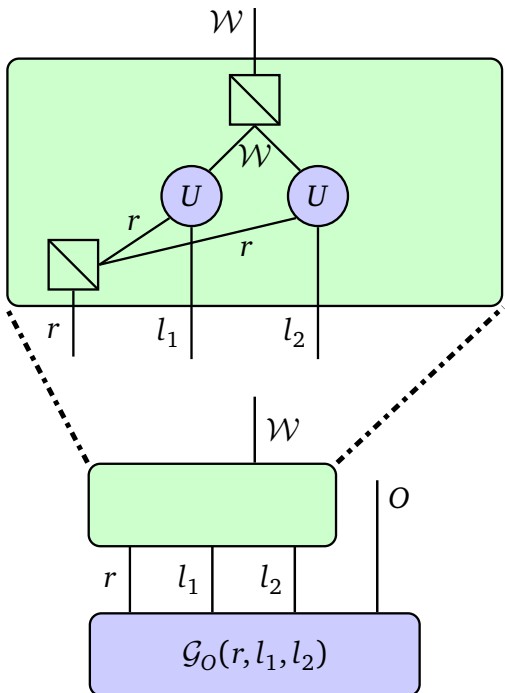

Figure 1: Graphical representation for Eq. (9). Each circle or rectangle with $N$ legs represents a $N$-way tensor. When two tensors share a leg, the summation over the corresponding index must be taken. A rectangle with a diagonal line represents an *identity* tensor. The $N$-way *identity* tensor $\boldsymbol{t}$ is defined as $t_{i_1 \cdots i_N} = 1$ iff $i_1 = \cdots = i_N$, and $t_{i_1 \cdots i_N} = 0$ otherwise.

Table 2: 12 different notations of Matsubara frequencies and statistics for four-point Green's functions in the particle-hole notation at a fixed bosonic frequency.

| $r$ | $(i\omega, i\omega')$ | $(\alpha, \alpha')$ |
|---|---|---|
| 1 | $(i\omega_n, i\omega_{n'})$ | $(F, F)$ |
| 2 | $(i\omega_n - i\omega_{n'}, i\omega_{n'})$ | $(\bar{B}, F)$ |
| 3 | $(i\omega_{n'} - i\omega_n, i\omega_n)$ | $(\bar{B}, F)$ |
| 4 | $(i\omega_n, i\omega_{n'} + i\nu_m)$ | $(F, F)$ |
| 5 | $(i\omega_n - i\omega_{n'}, i\omega_{n'} + i\nu_m)$ | $(\bar{B}, F)$ |
| 6 | $(i\omega_{n'} - i\omega_n, i\omega_n + i\nu_m)$ | $(\bar{B}, F)$ |
| 7 | $(i\omega_n + i\nu_m, i\omega_{n'})$ | $(F, F)$ |
| 8 | $(i\omega_n + i\omega_{n'} + i\nu_m, i\omega_{n'})$ | $(\bar{B}, F)$ |
| 9 | $(i\omega_{n'} + i\omega_n + i\nu_m, i\omega_n)$ | $(\bar{B}, F)$ |
| 10 | $(i\omega_n + i\nu_m, i\omega_{n'} + i\nu_m)$ | $(F, F)$ |
| 11 | $(i\omega_n + i\omega_{n'} + i\nu_m, i\omega_{n'} + i\nu_m)$ | $(\bar{B}, F)$ |
| 12 | $(i\omega_{n'} + i\omega_n + i\nu_m, i\omega_n + i\nu_m)$ | $(\bar{B}, F)$ |

imaginary-time frequencies $(i\omega, i\omega')$ and the statistics of the basis functions depend on $r$ as summarized in Table 2.

This expression is formally similar to Eq. (8), being meant to store the full information for a single bosonic frequency. The same sparse sampling strategy, which we shall introduce in Sec. 3, can therefore be employed in both situations.

## 2.4 Graphical representation

For the sake of clarity, we introduce a graphical representation for the tensor operations involving 2P Green's functions. As an example, we present in Fig. 1 the diagram corresponding to the right-hand side of Eq. (9). Each rectangle/circle represents a tensor whose indices are denoted by legs (the nature of the shape does not matter). The set of expansion coefficients $\mathcal{G}_O(r, l_1, l_2; i\nu_m)$ appears as a purple four-legged box, representing a rank four tensor. The green rectangle represents the basis functions terms in Eq. (9). A detailed view of their action is shown in the upper panel, where $\mathcal{W}$ is introduced as a composite index of $(i\omega, i\omega')$.

## 3 Sparse sampling

The sparse sampling scheme was originally proposed for a 1P Green's function in Ref. [23]. For fermions, the expansion of $G(i\omega_n)$ reads

$$G(i\omega_n) = \sum_{l=0}^{N_l-1} \mathcal{G}(l) U_l^F(i\omega_n), \tag{10}$$

where the number of coefficients $N_l = l_{\max} + 1$ determines the accuracy of the expansion. It was shown that the full frequency dependence of a 1P Green's function can be reconstructed from the values of the Green's function on a carefully chosen sparse subset of sampling points in the Matsubara domain. $U_l^F(i\omega_n)$ is real (odd $l$) or pure imaginary (even $l$), and oscillates around zero. The procedure described in Ref. [23] is based on picking the positions of the extrema of $|U_{l_{\max}}^F(i\omega_n)|$. The procedure generates $N_l$ (even $l$) or $N_l + 1$ (odd $l$) sampling points. The same procedure generates $N_l + 1$ (even $l$) or $N_l$ (odd $l$) sampling points for bosons.

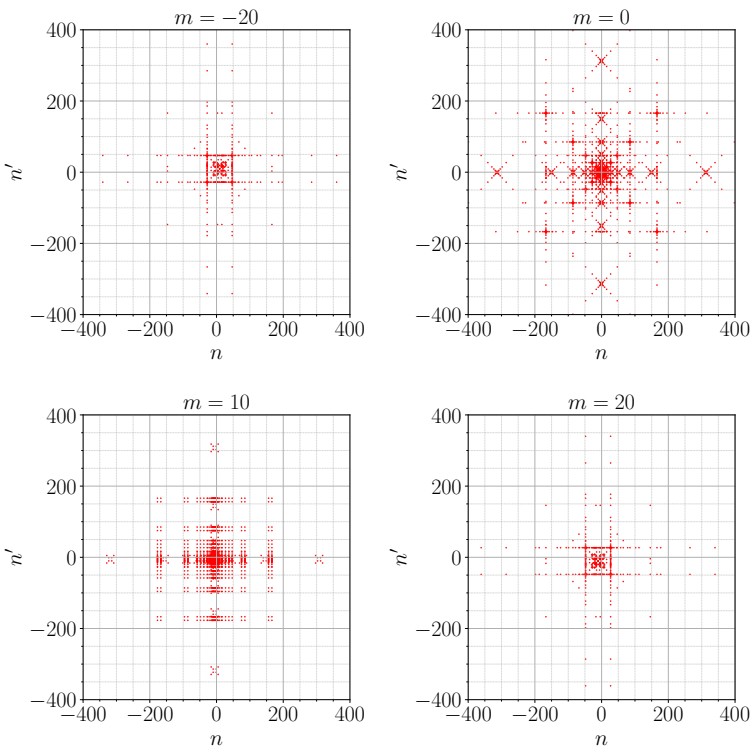

Figure 2: Sampling points generated in the three-frequency space for $\Lambda = 10^4$ and $N_l = 24$. We show cuts at $m = -20, 0, 10, 20$. We show only sampling points at low frequencies. The actual sampling points are distributed up to $|n|, |n'| \lesssim 1800$ and $|m| \lesssim 2200$.

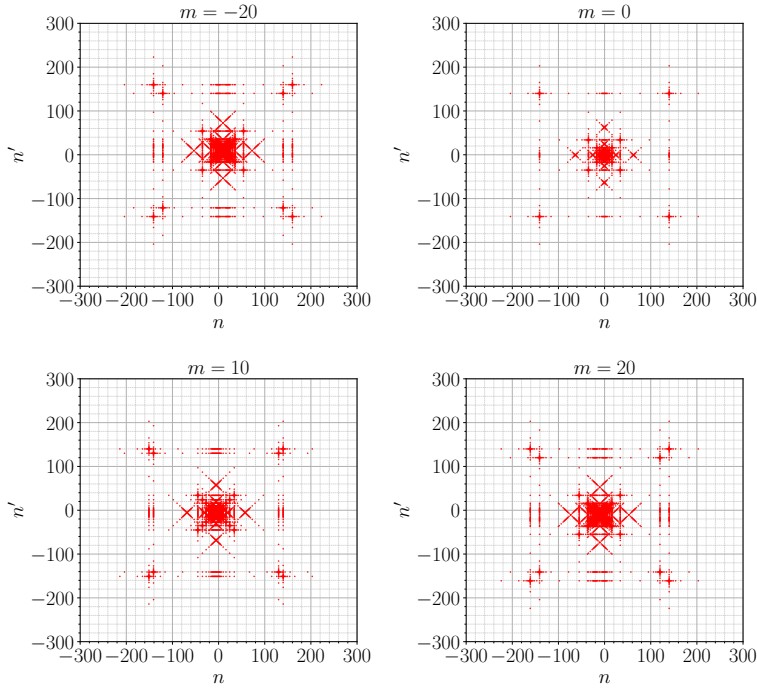

Figure 3: Sampling points generated in the particle-hole notation independently for $m = -20, 0, 10, 20$ ($\Lambda = 100$ and $N_l = 19$).

We extend this procedure in a straightforward manner for the expansion of the 2P Green's function. Each summand indexed by $r$ in Eq. (8) (Eq. (9) ) is handled in turn. For a fixed value of $r$, the sets of sampling points relative to each factor in the corresponding product of basis functions are built. Then, the triplets (pairs) of direct products of such sets are determined, and make up the $r^{\text{th}}$ set of points to be sampled, in $\mathbb{Z}^3$ ($\mathbb{Z}^2$).

For simplicity of implementation, we use the same $N_l$ for fermions and for bosons in the expansion. In general, for a given value of $\Lambda \equiv \beta\omega_{\max}$, the singular values decay slower for fermions than for bosons. In practice, we determine $N_l$ based on a given singular-value cutoff for fermions, and use the same $N_l$ for bosons.

As an illustration, for Eq. (8), we obtain $N_l^3 + O(1)$ sampling points. The final set of sampling points we need to consider is the union of the sets of sampling points obtained for all $r$. The size of the union is less than the sum of the sizes of the individual sets, thanks to the overlap between them (in particular at low frequencies). In Sec. 6, we will use $\Lambda = 10^4$ and $N_l = 24$ (cutoff value $10^{-4}$ for singular values). For this parameter set, the procedure generates $165\,912$ sampling points for Eq. (8), which is slightly smaller than $16N_l^3 = 221\,184$ due to the overlap. Figure 2 shows the distribution of these sampling points. One can see that their distribution is more dense at low frequencies, getting sparse at high frequencies.

Figure 3 shows the distribution of the sampling points generated for Eq. (9) with $N_l = 19$ (cutoff value of $10^{-5}$). We obtain $2\,972$, $1336$, $2\,516$, $2\,972$ sampling points for $m = -20, 0, 10, 20$, respectively. We will use these parameters in Sec. 5.

One technical caveat needs to be pointed out: the expansion of the 2P Green's functions involves the so-called "extended" bosonic basis set [22]. A basis function $U_l^{\overline{\text{B}}}$ from this set only exhibits $\max(0, l-2)$ sign changes, due to the extra basis functions at $l = 0, 1$. The procedure above would thus yield $l_{\max} - 1 = N_l - 2$ or $N_l - 1$ sampling points for this basis set. Therefore the actual process is slightly altered from the above description. The sampling points relative to the extended bosonic basis are generated from the extrema of $U_{l_{\max}+2}^{\overline{\text{B}}}$ instead of $U_{l_{\max}}^{\overline{\text{B}}}$. This ensures that the number of unknown coefficients matches the number of sampling points.

In the following sections, we will demonstrate that the sampling on the sparse grid is sufficient to evaluate the 2P Green's function with the desired precision for any Matsubara frequency.

# 4 Tensor network representation

In this section, we introduce an efficient fitting algorithm based on a tensor network representation for the IR tensor. We refer the reader who is not familiar with tensor networks to Refs. [26–28]. In principle, numerical data on the sampling points can be fitted using either Eq. (8) or Eq. (9) by using the least squares method. The computational load of this naive approach scales as $O(N_l^6 N_{\text{orb}}^4)$ or $O(N_l^9 N_{\text{orb}}^4)$ for two- and three-frequency quantities, respectively. Here $N_{\text{orb}}$ is the number of orbitals, and $N_l$ grows logarithmically with respect to $\beta$. The fitting rapidly becomes too costly at low temperature, especially for three-frequency quantities.

## 4.1 Low-rank tensor decomposition

We introduce the following low-rank decomposition of $G$:

$$\mathcal{G}_O(r, l_1, l_2, l_3) \simeq \tilde{\mathcal{G}}_O(r, l_1, l_2, l_3) \equiv \sum_{d=1}^{D} x_{dr}^{(0)} x_{dl_1}^{(1)} x_{dl_2}^{(2)} x_{dl_3}^{(3)} x_{dO}^{(4)}, \tag{11}$$

where $r$ runs over different representations and $O \equiv (i, j, k, l)$. This type of tensor decomposition is known as a Canonical Polyadic (CP) decomposition and is widely used in many fields,

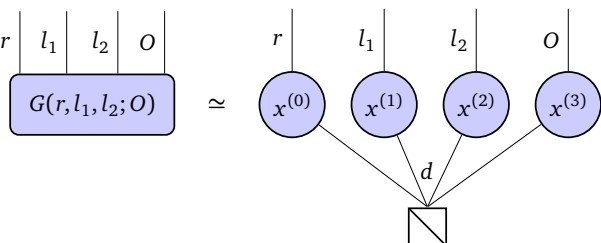

Figure 4: Graphical representation of the tensor decomposition in Eq. (12). Each circle or rectangle with $N$ legs represents a $N$-way tensor. When two tensors share a leg, the summation over the corresponding index must be taken. A rectangle with a diagonal line represents a *identity tensor*.

e.g., for accelerating quantum chemistry calculations by factorizing Coulomb integrals [29–31]. For the simplified form with fixed bosonic frequency in Eq. (9), the low-rank decomposition reads

$$\mathcal{G}_O(r, l_1, l_2; i\nu_m) \simeq \tilde{\mathcal{G}}_O(r, l_1, l_2; i\nu_m) \equiv \sum_{d=1}^{D} x_{dr}^{(0)}(\nu_m) x_{dl_1}^{(1)}(\nu_m) x_{dl_2}^{(2)}(\nu_m) x_{dO}^{(3)}(\nu_m), \tag{12}$$

which is illustrated in Figure 4.

Both expressions become exact for a large enough $D$. The decomposition is beneficial for the fitting procedure if a good approximation of the full tensor is obtained for a reasonably small value of $D$. We discuss further in the text how this condition can be checked numerically. Note that the dependence on the orbital indices is not decomposed and they still appear as the composite index $O$ in Eqs. (11) and (12). In the CP decomposition, we do not assume any orthogonality conditions for the decomposed tensors, unlike the SVD of a matrix. Recently, some of the authors and co-workers have proposed strong-coupling formulas for computing momentum dependent susceptibilities in DMFT [32]. Their formulas can be regarded as a special case of Eq. (12) with $D = 1$ and $x_{dr} = \delta_{r1}$. Our preliminary results indicate that the further decomposition of the dependence on $O$ into individual spin/orbital indices requires typically an even larger $D$, which is not beneficial.

## 4.2 Fitting algorithm

We explain how to fit some existing data of the 2P Green's function on the sampling points in the Matsubara domain using Eq. (11) or Eq. (12) without explicitly constructing the big tensors on the left-hand side. The $x$ tensors in the equation can be regarded as free fitting parameters. For instance, for Eq. (11), we define a cost function for the fitting as

$$f(\{x^{(i)}\}) = ||G(\mathcal{W}, O) - \tilde{G}(\mathcal{W}, O)[x^{(i)}]||_2^2 + \alpha \sum_{i=0}^{4} ||x^{(i)}||_2^2, \tag{13}$$

where $||\ldots||_2$ denotes the Frobenius norm and $\mathcal{W}$ runs over sampling points in the Matsubara frequency domain. $G(\mathcal{W}, O)$ are the data of the Green's function on the sampling points which we fit, while $\tilde{G}(\mathcal{W}, O)[x^{(i)}]$ denotes the data of the Green's function evaluated from $\{x^{(i)}\}$. We introduce the small parameter $\alpha > 0$ in order to regularize the optimization problem. Without this parameter, the problem would be ill-posed due to the overcompleteness of the representation. We have not observed any visible systematic errors in interpolated data for small values of $\alpha$, i.e., $\alpha \leq 10^{-5}$. We used $\alpha = 10^{-8}$ and $10^{-5}$ in Sec. 5 and Sec. 6, respectively.

The minimization of this cost function is a non-convex optimization problem. We found that despite its non-convex nature, the cost function can be minimized efficiently using standard methods starting from randomly initialized parameters. In some cases, we observed the existence of multiple solutions, being different only slightly in terms of the cost function. This issue thus does not matter in practice. We refer the interested reader to Appendix B for more details on the optimization method.

# 5 DMFT calculations for single-band Hubbard model on a square lattice

As a test bed for our method, we first consider a single-band Hubbard model on a square lattice at half filling for $U = 12t$ ($1.5\times$ the bandwidth $W = 8t$), where the hopping $t = 1$ sets the unit of energy. The inverse temperature $\beta = 2.5$ is slightly above the antiferromagnetic transition. We use (approximate) Hubbard-I solver, which provides semi-analytic representation of local susceptibilities and thus allows precise analysis of our data compression approach. We first compute and fit the local (impurity) generalized susceptibility $X^{\text{loc}}$ by subtracting the relevant disconnected parts from the local 2P Green's function. Interpolating the local generalized susceptibility in the Matsubara frequency domain, we solve the Bethe-Salpeter equation (BSE) and compute the static DMFT susceptibilities of the model.

We compute $X^{\text{loc}}$ for all spin-orbital components on 1 336 sampling points generated for $\Lambda = 100$ and $N_l = 19$ at zero bosonic frequency $m = 0$. The distribution of the sampling points is shown in the right top panel of Fig. 3. Then, we fit the data using Eq. (12) ($G$ is simply replaced by $X^{\text{loc}}$). Figure 5 shows how the fitting errors decay as $D$ is increased. We found that the residual of the fit vanishes quickly with increasing $D$. Figure 6 compares the exact and interpolated values of the local susceptibility in Matsubara frequency space. One can see that for $D = 15$, the fit matches the exact values on the sampling points and precisely interpolates the data. Increasing $D$ further does not improve the fit substantially, which may be due to the truncation of the basis.

The inversion of the Bethe-Salpeter equations (for the determination of the lattice susceptibilities) directly in the IR and tensor network format is still an open question (see the discussion in Sec. 7). In this study, we execute the inversion itself using the Matsubara representation, based on the interpolation of the generalized susceptibility in a box of width $[-N_\omega, N_\omega - 1]$ for fermionic frequencies. Corrections from higher frequencies outside the box are treated using the procedure described in Appendix B of Ref. [32]. The corresponding physical quantities are obtained by summation over the fermionic frequencies.

Figure 7 shows the physical lattice susceptibility using the fitted results for $D = 5$ and $i\nu = 0$. For the calculations in this section, we took $N_\omega = 100$. We found that the results for $D = 5$ are already indistinguishable from the exact ones at the scale of the figure. There is a pronounced peak at $M = (\pi, \pi)$, corresponding to an antiferromagnetic spin order. The other three eigenvalues, which correspond to charge susceptibility, are not enhanced.

The storage of $X^{\text{loc}}$ computed on the sampling points takes up 340 kB (including all spin sectors), while the compressed data for $D = 5$ only 5.2 kB.

# 6 DMFT calculations for two-band Hubbard model

Next, we apply the present IR approach to a state-of-art linear response DMFT calculation. To this end, we choose the two-band Hubbard model in an intermediate coupling regime and low spontaneously broken symmetry, a problem that some of us have studied recently [33]. The

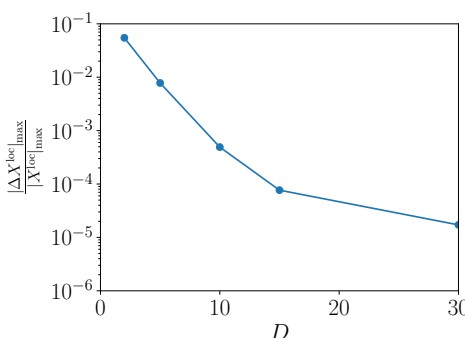

Figure 5: Fitting errors of $X^{\mathrm{loc}}$ for the single-band Hubbard model on the square lattice at $U = 1.5W$ and $\beta = 2.5$.

model Hamiltonian reads

$$H = \sum_{ij,\sigma}\Big(t_a a_{i\sigma}^\dagger a_{j\sigma} + t_b b_{i\sigma}^\dagger b_{j\sigma}\Big) + \frac{\Delta}{2}\sum_{i,\sigma}(n_{i\sigma}^a - n_{i\sigma}^b) + U\sum_{i,\alpha} n_{i\uparrow}^\alpha n_{i\downarrow}^\alpha + \sum_{i,\sigma\sigma'}(U' - J\delta_{\sigma\sigma'})n_{i\sigma}^a n_{i\sigma'}^b,$$

where $a_{i\sigma}^\dagger$ and $b_{i\sigma}^\dagger$ are fermionic operators that create electrons with the respective orbital flavors and spin $\sigma$ at site $i$ of a square lattice. The first term describes the nearest neighbor hopping. The rest, expressed in terms of local densities $n_{i,\sigma}^c \equiv c_{i\sigma}^\dagger c_{i\sigma}$, captures the crystal-field $\Delta$, the Hubbard interaction $U$ and Hund's exchange $J$ in the Ising approximation. We use the same hopping parameters $t_a = 0.4118$, $t_b = -0.1882$ as in [33], but choose weaker interaction $U = 2$ ($J = U/4$, $U' = U - 2J$) and $\Delta = 1.55$. At the sstudied inverse temperature $\beta = 60$, an ordered phase called polar excitonic condensate [34] is realized. We follow the same the algorithmic programme as in Ref. [33], except for the representation of the 2P Green's function.

The local 2P Green's function $G^{\mathrm{loc}}$ is sampled on a non-uniform grid in the Matsubara frequency domain, shown in Fig. 2 ($\Lambda = 10^4$, $N_l = 24$), using a modified version of the ALPS/CT-HYB impurity solver [35,36] based on the continuous-time hybridization expansion algorithm [37, 38]. The regression (11,13) for $G^{\mathrm{loc}}$ provides us with the $\overline{x}^{(i)}$, $i \in \{0,\ldots,4\}$ coefficients, Eq. (11), which in turn are used to interpolate the local 2P Green's function at all Matsubara frequencies. The generalized local susceptibility $X^{\mathrm{loc}}$ is then evaluated by subtracting the disconnected part from the local 2P Green's function.

Figure 8 illustrates the convergence of the process with increasing value of $D$ in this situation. The solid lines represent the real part of $G^{\mathrm{loc}}$ in the Matsubara representation, obtained from Eq. 8. Both panels show the profile for a fixed bosonic frequency in the particle-hole notation ($m = 0$ and $m = 10$), along a cut in the two-dimensional fermionic frequency space, slightly away from the main diagonal (shifted by ten Matsubara frequencies). The crosses represent the actual sparse frequencies, where the data was sampled. The structure of the data at zero bosonic frequency is relatively simple, so that the fit is excellent for $D = 30$, but the tenth bosonic frequency requires $D = 60$.

Using the above data we can solve the BSE to get the linear response of our system. As an

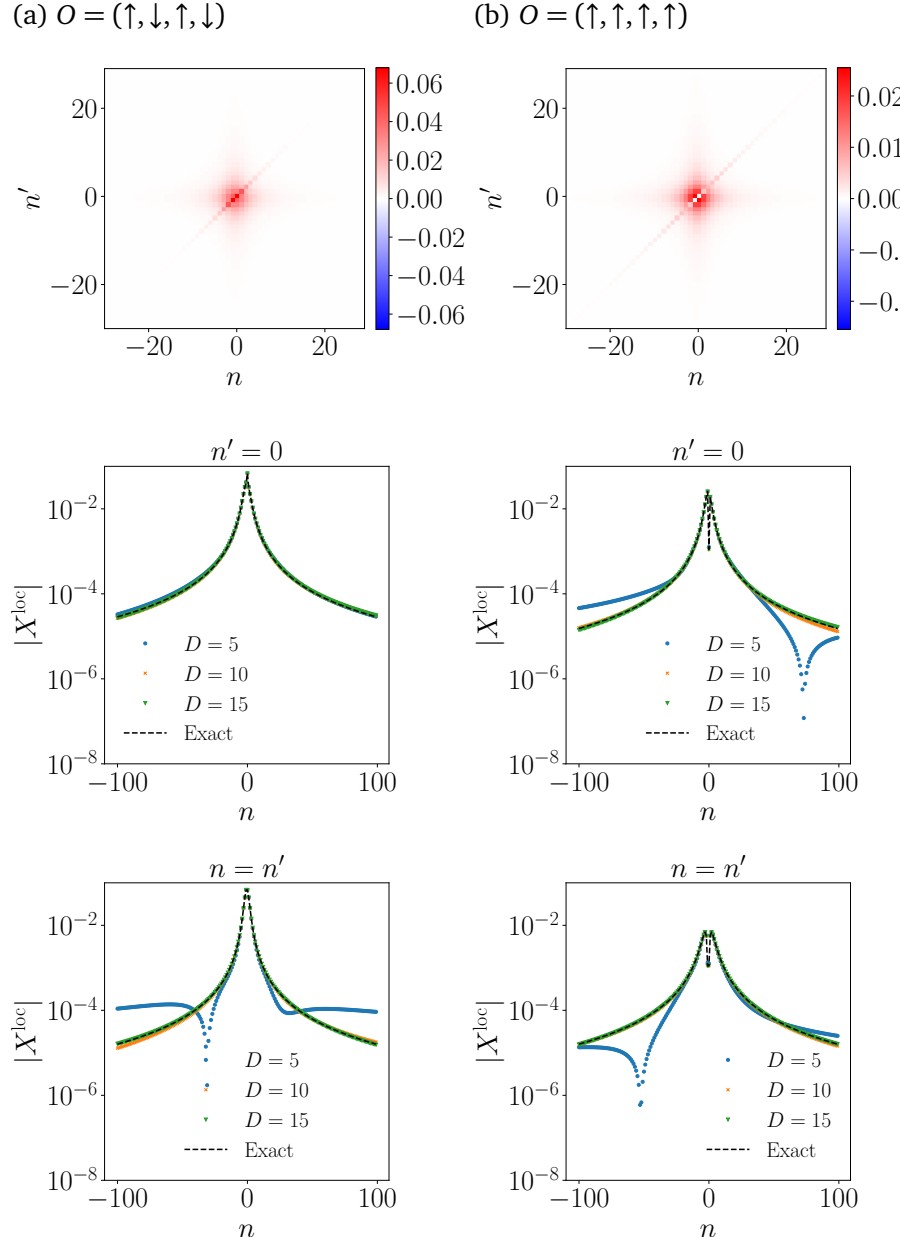

Figure 6: Comparison of interpolated results and exact values of the generalized susceptibility $X^{\text{loc}}$ for the single-band Hubbard model. The panels (a) and (b) show the data for $O = (\uparrow, \downarrow, \uparrow, \downarrow)$ and $(\uparrow, \uparrow, \uparrow, \uparrow)$, respectively. The 2D colormaps show the interpolated data for $D = 15$.

example we calculate the diagonal susceptibilities for local operators [33]

$$S_z = \sum_{c=a,b} \left( n_\uparrow^c - n_\downarrow^c \right),$$

$$\mathcal{R}\phi_\gamma = \sum_{\alpha\beta} \sigma_{\alpha\beta}^\gamma \left( a_\alpha^\dagger b_\beta + b_\alpha^\dagger a_\beta \right),$$

$$\mathcal{I}\phi_\gamma = \sum_{\alpha\beta} i\sigma_{\alpha\beta}^\gamma \left( a_\alpha^\dagger b_\beta - b_\alpha^\dagger a_\beta \right),$$

which capture the low-energy dynamics of the polar condensate. We have chosen the ordered

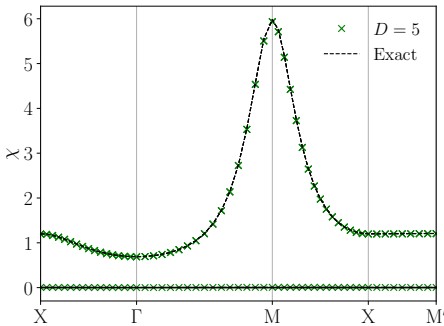

Figure 7: Physical lattice susceptibility $\chi(q)$ computed for the single-band Hubbard model on the square lattice at $U = 1.5W$ and $\beta = 2.5$. We used $D = 5$. Different branches correspond to different eigen modes of $\chi(\boldsymbol{q}, 0)$, i.e., the spin susceptibility and the charge susceptibility.

(a) $O = (a\uparrow, a\uparrow, a\uparrow, a\uparrow), m = 0$

(b) $O = (b\downarrow, b\downarrow, a\uparrow, a\uparrow), \text{m} = 10$

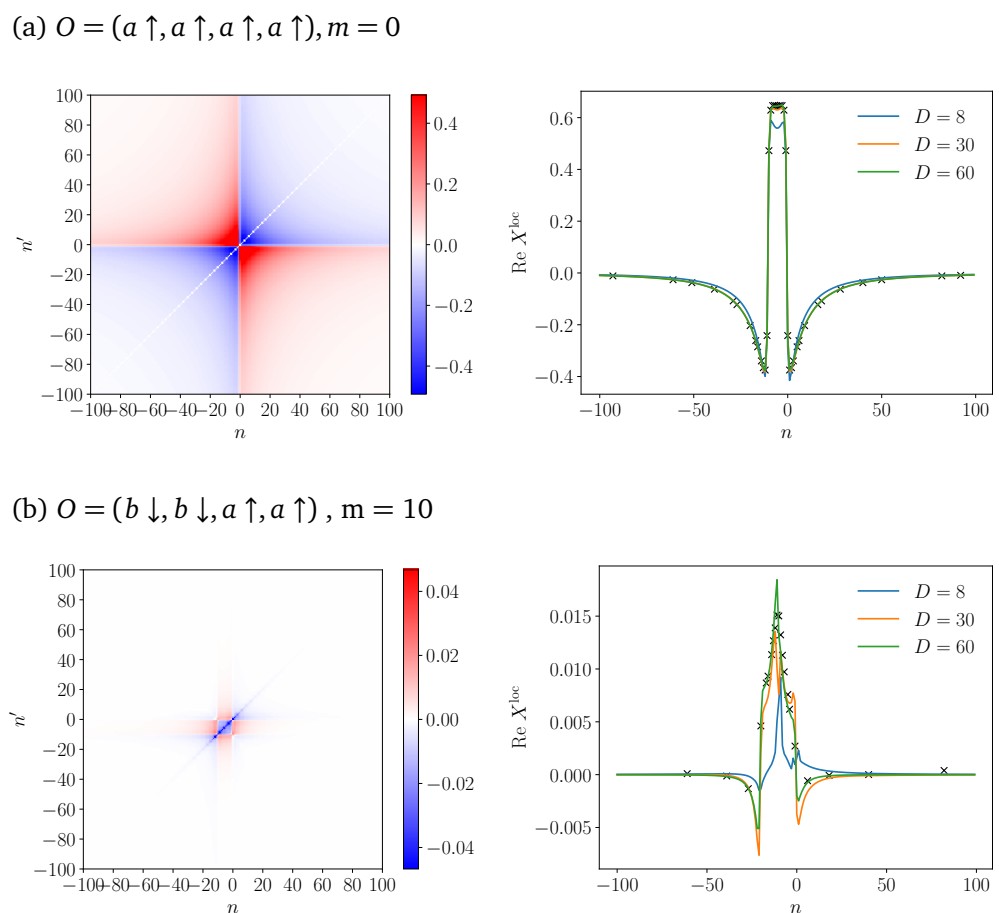

Figure 8: Local 2P Green's function $G^{\text{loc}}$ interpolated at zero bosonic frequency $m = 0$ [(a)] and a finite bosonic frequency $m = 10$ [(b)] for the two-band Hubbard model. The 2D color maps show results of the local 2P Green's function for $D = 60$. The right-hand halves of the panels (a) and (b) show the convergence of the interpolated values with respect to $D$ on the line of $n' = n + 10$.

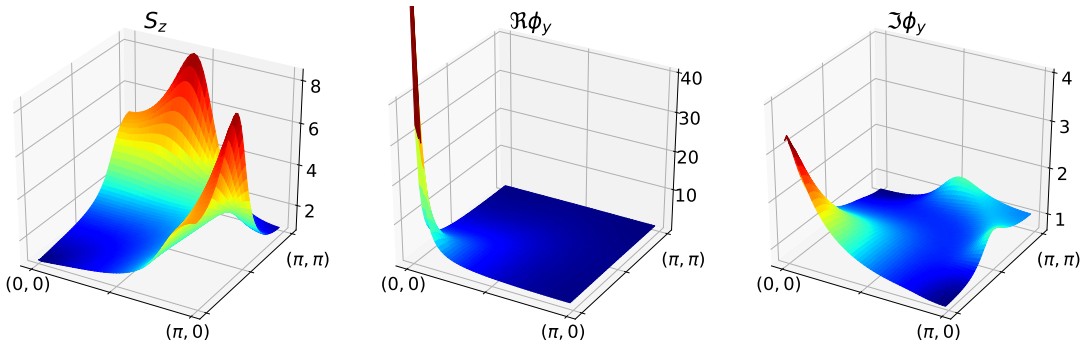

Figure 9: Selected components of the static susceptibility $\chi(\boldsymbol{q})$ throughout the Brillouin zone, for the two-band Hubbard model described in the text, for $\beta = 60$.

phase such that only $\langle \mathcal{R}\phi_x \rangle$ is non-zero. In this set-up $\mathcal{R}\phi_y$ generates a spin rotation of the order parameter (Goldstone mode) and $\mathcal{I}\phi_y$ couples to $S_z$ (only in the ordered phase).

In Fig. (9) we show the static susceptibilities on a fine grid in the 2D Brillouin zone. Unlike in the strong coupling case of Ref. [33], the spin susceptibility is dominated by a Fermi surface nesting present both in the ordered and disordered (not shown here) phases, which gives rise to a peak at an incommensurate vector on the $X$-$M$ zone boundary. The response corresponding to the Goldstone mode in the middle panel exhibits the expected divergence at the ordering wave vector ($\mathbf{q} = 0$). The excitonic susceptibility in the right panel exhibits, in addition to the main (finite) peak at $\mathbf{q} = 0$, additional peaks that coincide with the maxima of the spin susceptibility. This reflects the coupling between $S_z$ and $\mathcal{I}\phi_y$ induced by the symmetry breaking. While in the strong coupling regime of Ref. [33] $S_z$ was a slave to the dynamics of $\mathcal{I}\phi_y$, here we can see that $S_z$ affects $\mathcal{I}\phi_y$. In Fig. (10) we show the absorptive (imaginary) parts of the dynamical susceptibilities obtained by the analytic continuation described in Supplemental Material of Ref. [33]. It reveals the Goldstone nature of the $\mathcal{R}\phi_y$ response and complex nature of the spin response. Interestingly, the sharp response in the vicinity of $\Gamma$ does not reflect formation of a bound state, but is a consequence of parallel bands upon opening of the excitonic gap. The low-energy hot spot on the $X$-$M$ linearand its counterpart in the static susceptibility reflect the vicinity of an antiferromagnetic phase [39].

The presented susceptibilities obtained from the IR inputs are in excellent agreement with benchmark data obtained using the Legendre representation used in Ref. [33]. The current setup is more flexible, insofar as it is not limited by the sampling window, neither in the bosonic nor the fermionic Matsubara frequency domain. It is also very compact. The sparse grid solely saves computational time and memory footprints for QMC significantly. The tensor regression further compresses the sparsely sampled data by several orders of magnitude: The measured data is 700 MB large on the sparse grid, while the tensor network representation takes up only 330 kB for $D = 60$.

In general, if a too large $D$ is employed, one could overfit QMC noise, giving a rise to oscillatory behavior between the sampling points in the interpolated data. A practical recipe for avoiding overfitting is to use the value of $D$ that minimizes test errors rather than training errors. In the present study, however, we did not observe overfitting behavior. This may be because the fitting parameters is still much smaller than the fitted QMC data in size. More detailed analysis of the stability of the fitting procedure is a topic of future studies.

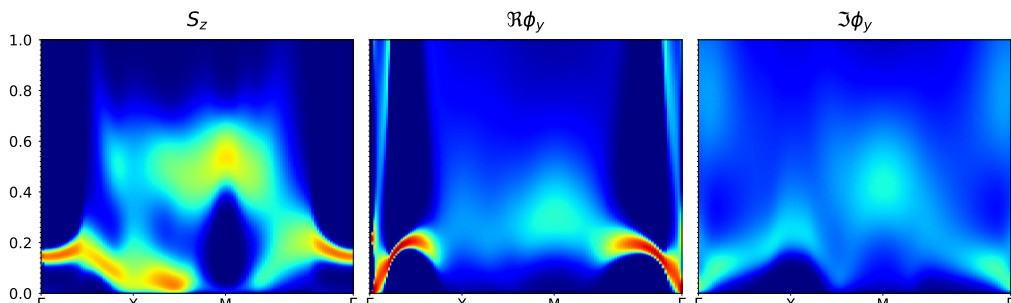

Figure 10: Selected modes of the dynamical susceptibilityy for the two-band Hubbard model described in the text, in the presence of the excitonic condensate, $\beta = 60$.

# 7 Summary

Based on the IR basis, we have introduced a procedure for generating sparse grids in the Matsubara frequency domain and a fitting algorithm based on a tensor network representation. These two enable an efficient transformation of numerical data from Matsubara to IR domain. The tensor network representation provides a model-independent way to compress the IR expansion coefficients (IR tensor) by decoupling the frequency and spins/orbital dependence. Low-temperature calculations for multi-orbital systems benefit from this compression.

We have demonstrated the efficiency and accuracy of the present method in DMFT calculations: static susceptibility calculations for single-band Hubbard model and dynamic susceptibility calculations for two-band Hubbard model with low symmetry. We have shown that accurate susceptibilities can be obtained already with low-rank approximation of the IR tensor.

The sparse sampling and the tensor network decomposition are independent procedures that are controlled separately. The size of sparse sampling grid, and thus its computational cost, depends only on temperature, the energy window and the desired accuracy. The "compression rate" and the accuracy of tensor network representation are controlled by the rank of decomposition $D$. In the present work, we have demonstrated that the local 2P Green's function can be compressed from 700 MB to 330 kB for the two-band Hubbard model. The concept of tensor network representation is flexible and further compression may be possible for different tensor network topologies. The choice of ideal tensor network topology requires an extensive experience with the performance of the method for various models and is beyond the scope of the present work.

Potential applications of the present scheme include DFT+DMFT calculations for realistic multi-orbital models and diagrammatic extensions of DMFT. It is highly desirable to develop efficient methods for solving equations at the 2P level such as Bethe-Salpeter and parquet equations directly in the tensor network format with the sparse sampling. This requires efficient evaluation of contractions of 2P quantities, e.g., a vertex function and a generalized susceptibility. Potentially useful techniques for manipulating matrix product states and tensor networks have already been developed in other fields of condensed matter theory [26,27,40].

# Acknowledgments

HS thanks Lei Wang for stimulating discussions on tensor networks and machine learning. Part of the calculations were run on the facilities of the Supercomputer Center at the Institute for Solid State Physics, University of Tokyo, using codes based on ALPSCore [41, 42]. We used the irbasis library [43] for computing IR basis functions. We used DCore [44] based

on TRIQS [45] and TRIQS/DFTTools [46] for DMFT calculations of the single-band Hubbard model.

**Funding information** H.S, J.O and K.Y were supported by JSPS KAKENHI Grant No. 18H01158. H.S. was supported by JSPS KAKENHI Grant No. 16K17735. J.O. was supported by JSPS KAKENHI Grant No. 18H04301 (J-Physics). K.Y. was supported by Building of Consortia for the Development of Human Resources in Science and Technology, MEXT, Japan. D.G and J.K were supported by the ERC Grant Agreements No. 646807 under EU Horizon 2020. D.G was supported by the Czech Science Foundation (GAČR) under Project No. GA19-16937S. This work was supported by The Ministry of Education, Youth and Sports from the Large Infrastructures for Research, Experimental Development and Innovations project "IT4Innovations National Supercomputing Center – LM2015070". MW and EG were supported by the Simons Foundation via the Simons Collaboration on the many-electron problem. Access to computing and storage facilities owned by parties and projects contributing to the National Grid Infrastructure MetaCentrum provided under the programme "Projects of Large Research, Development, and Innovations Infrastructures" (CESNET LM2015042), and by the Austrian Federal Ministry of Science, Research and Economy through the Vienna Scientific Cluster (VSC) Research Center, is greatly appreciated.

# A   Intermediate representation at fixed bosonic frequency

The frequency dependence of $G(i\omega_n, i\omega_{n'}, i\nu_m)$ can be decomposed into 16 distinct components shown in Table. 1. For instance, the first component ($r = 1$) depends on the three frequencies through $i\nu_m + i\omega_n, -i\omega_n, i\omega_{n'}$, which defines the structure of discontinuity planes in the imaginary-time domain. To be more specific, as discussed in Ref. [22], the first component can be discontinuous at three equal-time planes: $\tau_1 = \tau_4$, $\tau_2 = \tau_4$ and $\tau_3 = \tau_4$ (mod $\beta$). Such a function with this discontinuity structure may be well approximated by

$$\frac{\rho^{(1)}(\epsilon_1, \epsilon_2, \epsilon_3)}{(i\nu_m + i\omega_n - \epsilon_1)(-i\omega_n - \epsilon_2)(i\omega_{n'} - \epsilon_3)}, \tag{14}$$

where $\rho^{(1)}(\epsilon_1, \epsilon_2, \epsilon_3)$ is an *auxiliary* spectrum bounded in $[-\omega_{\max}, \omega_{\max}]$. Applying the same procedure to all the 16 components, we obtain the assumption that

$$G(i\omega_n, i\omega_{n'}, i\nu_m) \simeq \sum_{r=1}^{16} \int_{-\omega_{\max}}^{\omega_{\max}} d\epsilon_1 d\epsilon_2 d\epsilon_3 \frac{\rho^{(r)}(\epsilon_1, \epsilon_2, \epsilon_3)}{(i\omega - \epsilon_1)(i\omega' - \epsilon_2)(i\omega'' - \epsilon_3)}. \tag{15}$$

For instance, the term for $r = 1$ can be decomposed as

$$\begin{aligned}
&\frac{1}{(i(\nu_m + \omega_n) - \epsilon_1)(-i\omega_n - \epsilon_2)(i\omega_{n'} - \epsilon_3)} \\
&= \frac{1}{i\nu_m - \epsilon_1 - \epsilon_2} \left( \frac{1}{i(\nu_m + \omega_n) - \epsilon_1} + \frac{1}{-i\omega_n - \epsilon_2} \right) \frac{1}{(i\omega_{n'} - \epsilon_3)} \\
&= \frac{1}{i\nu_m - \epsilon_1 - \epsilon_2} \left\{ K^{\mathrm{F}}(i\omega_n + i\nu_m, \epsilon_1) - K^{\mathrm{F}}(i\omega_n + \epsilon_2) \right\} K^{\mathrm{F}}(i\omega_{n'}, \epsilon_3).
\end{aligned} \tag{16}$$

One can see the first and second terms in the last line are the products of two fermionic kernels. The first term can be represented compactly as

$$\sum_{l_1, l_2} \left( \frac{S_{l_1}^{\mathrm{F}} S_{l_2}^{\mathrm{F}} V_{l_1}^{\mathrm{F}}(\epsilon_1) V_{l_2}^{\mathrm{F}}(\epsilon_3)}{i\nu_m - \epsilon_1 - \epsilon_2} \right) U_{l_1}^{\mathrm{F}}(i\omega_n + i\nu_m) U_{l_2}^{\mathrm{F}}(i\omega_{n'}), \tag{17}$$

where the coefficient in the parenthesis decays as fast as the singular values with respect to $l_1$ and $l_2$.

Applying the same procedure to all the terms in Eq. (15), one obtains a compact overcomplete representation

$$G(i\omega_n, i\omega_{n'}, i\omega_m) = \sum_{s,s'=0,1}\sum_{l_1,l_2=0}^{\infty} \left\{ \mathcal{G}^{(1)}_{ss'l_1l_2} U^{\mathrm{F}}_{sl_1}(i\omega_n) U^{\mathrm{F}}_{s'l_2}(i\omega_{n'}) \right.$$
$$+ \mathcal{G}^{(2)}_{ss'l_1l_2} U^{\overline{\mathrm{B}}}_{l_1 s}(i\omega_n + (-1)^{s+1}i\omega_{n'}) U^{\mathrm{F}}_{l_2 s'}(i\omega_{n'})$$
$$+ \left. \mathcal{G}^{(3)}_{ss'l_1l_2} U^{\overline{\mathrm{B}}}_{l_1 s}(i\omega_{n'} + (-1)^{s+1}i\omega_n) U^{\mathrm{F}}_{l_2 s'}(i\omega_n) \right\}. \tag{18}$$

Here we defined

$$U^{\alpha}_{sl}(i\omega_n) \equiv U^{\alpha}_l(i\omega_n + si\omega_m) \tag{19}$$

for $s = 0, 1$.

We can recast Eq. (18) into a form more analogous to Eq. (8) as

$$G(i\omega_n, i\omega_{n'}, i\omega_m) = \sum_{r=1}^{12}\sum_{l_1,l_2} \mathcal{G}(r, l_1, l_2; i\nu) U^{(r)}_{l_1}(i\omega) U^{(r)}_{l_2}(i\omega'). \tag{20}$$

# B   Optimization algorithm for tensor regression

We minimize the cost function in Eq. (13) by means of a accelerated alternating least squares (ALS) method. The essential idea of ALS is to optimize each tensor in $\{x^{(0)}, x^{(1)}, \cdots\}$ at one time. The optimization of a single tensor reduces to a convex optimization problem. In ALS, we sweep through all the tensors until the value of the cost function is converged. In addition, we introduce recently proposed acceleration techniques to improve the convergence of ALS. In the following, we detail the procedure of ALS and the acceleration techniques.

## B.1   Alternating least squares

We explain how to optimize the tensors in Eq. (13) by alternating least squares.

### B.1.1   Optimization of $x^{(0)}$

The minimization of Eq. (13) with respect to $x^{(0)}$ can be recast into

$$\min_{x^{(0)}} ||y(W, O) - \sum_{r,d} A(W, O; d, r) x^{(0)}_{dr}||^2 + \alpha ||x^{(0)}||^2, \tag{21}$$

which is a regularized convex optimization in the well-known form of Ridge regression. The tensor $A$ reads

$$A(W, O; d, r) \equiv B(W, r, d) x^{(3)}_{dO}, \tag{22}$$

where

$$B(W, r, d) \equiv \sum_{l_1} U^{(r)}_{l_1}(i\omega) x^{(1)}_{dl_1} \sum_{l_2} U^{(r)}_{l_2}(i\omega') x^{(2)}_{dl_2}. \tag{23}$$

In matrix form, Eq. (21) reads

$$\min_{x^{(0)}} ||y - Ax^{(0)}||^2 + \alpha||x^{(0)}||^2, \tag{24}$$

where $y$ and $x^{(3)}$ are flatted 1D arrays. The matrix is size of $(N_W N_O, N_r D)$, where $N_r$ is the number of different representations (=12), $N_O$ is the size of the combined index for spin and orbitals.

In practice, we solve Eq. (24) by an iterative method, LSQR [47], without constructing the matrix $A$ explicitly. In LSQR, the matrix $A$ is used only to compute $Av$ and $A^\dagger u$ for various $v$ and $u$. Hence we store the (precomputed) tensor $B$ in memory, and compute these products by means of tensor contractions. We illustrate the tensor contractions for computing $Av$ and $A^\dagger u$ in Figs. 11(a) and 11(b), respectively. This approach not only reduces memory footprints but also reduces the computational complexity from $O(N_W D N_r N_O)$ to $O(N_W D N_r) + O(N_W D N_O)$.

### B.1.2   Optimization of $x^{(1)}$

The optimization of $x^{(1)}$ can be done in a way very similar to that of $x^{(0)}$. Thus, we focus only on the differences. The reduced least squares problem reads

$$\min_{x^{(1)}} ||y(W, O) - \sum_{d,l_1} A'(W, O; d, l_1)x^{(1)}_{dl_1}||^2 + \alpha||x^{(1)}||^2, \tag{25}$$

where

$$A'(W, O; d, l_1) \equiv B'(W, d, l_1)x^{(1)}_{dO}, \tag{26}$$

where

$$B'(W, d, l_1) \equiv x^{(0)}_{dr} \sum_{l_1} U^{(r)}_{l_1}(i\omega)x^{(1)}_{dl_1}. \tag{27}$$

As illustrated in Fig. 11, we can readily compute $A'v$ and $A'^\dagger u$ by tensor contractions.

### B.1.3   Optimization of $x^{(2)}$

The tensor $x^{(2)}$ can be optimized in exactly the same way as $x^{(1)}$. Thus, we do not describe the optimization of $x^{(2)}$ for simplicity.

### B.1.4   Optimization of $x^{(3)}$

The optimization of $x^{(3)}$ is rather simple. The reduced least squares problem reads

$$\min_{x^{(3)}} ||y(W, O) - \sum_{d,O} A''(W; d, O)x^{(3)}_{dO}||^2 + \alpha||x^{(3)}||^2, \tag{28}$$

where the tensor $A''$ can be stored in memory. Furthermore, this optimization problem is separable with respect to $D$ and thus can be solved independently.

## B.2   Acceleration techniques and convergence condition

In the previous subsection, we have explained how to perform one sweep through the tensors. This single ALS sweep corresponds to the function ALS in Algorithm 1. The function ALS takes an array obtained by flattening tensors of fitting parameters as input. After a single sweep, the updated tensors are returned as a flattened array. Here, flattening means recasting multiple

tensors of complex numbers into a single one-dimensional array of real numbers (the order is arbitrary).

The whole procedure of the accelerated ALS is illustrated in Algorithm 1. The main difference from the plain ALS is the existence of $\beta$. In the second last line, $\beta(\boldsymbol{x}_k - \boldsymbol{x}_{k-1})$ acts as a momentum term for $\beta > 0$. Although this momentum term accelerates the convergence by updating the parameters aggressively, this sometimes leads to oscillatory behavior or divergence. We use a restarting mechanism to stabilize the accelerated ALS. In practice, when the restarting condition $f(\boldsymbol{x}_k) > f(\boldsymbol{x}_{k-1})$ is met ($f$ is the cost function), a ALS sweep is forced by setting $\beta = 0$ (see the comment in Algorithm 1). For more details on the acceleration techniques, please refer to Ref. [48].

The loop is exited when a convergence condition is met. $\delta$ in Algorithm 1) is a relative tolerance.

## B.3 Technical details and numerical results

We parallelize the whole fitting procedure by MPI with respect to frequencies. This parallelization is efficient particularly for $G$. We parallelize the LSQR implementation in the SciPy Python packageusing MPI with respect to sampling frequencies.

Figure 12 shows the convergence of the root squared errors of the local susceptibility for the single-band Hubbard model analyzed in Sec. 5. One can see that the fitting errors quickly converge. The small oscillatory behavior is due to the acceleration.

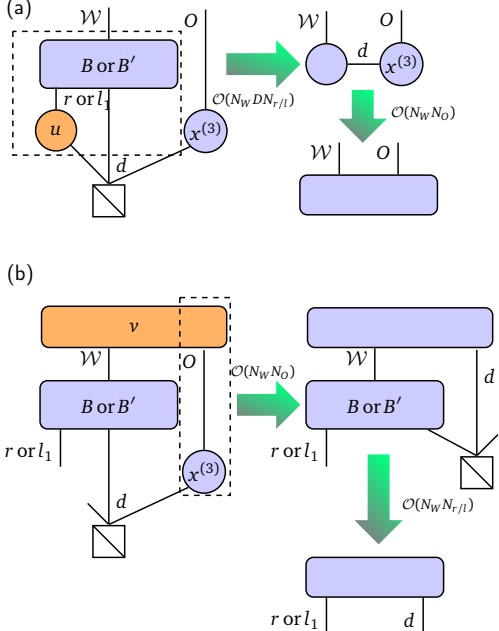

Figure 11: Graphical representations of tensor contractions for solving Eqs. (21) and (25), iteratively. The panel (a) illustrates the order of tensor contractions for computing $A\boldsymbol{u}$ and $A'\boldsymbol{u}$. The panel (b) illustrates the procedure for computing $A^\dagger \boldsymbol{v}$ and $(A')^\dagger \boldsymbol{v}$. The computational complexity for each operation is shown.

---

**Algorithm 1** Accelerated alternating least squares

---
**function** ALS($x$)
    $\{x^{(i)}\} \leftarrow$ UNFLATTEN($x$)                                    ▷ Unflatten input array
    **for** $i = 0, 1, 2, 3$ **do**
        Construct $A^{(i)}$
        $x^{(i)} \leftarrow$ LSQR($y, A^{(i)}, \alpha$)
    **end for**
    Return FLATTEN($\{x^{(i)}\}$)                                    ▷ Flatten updated tensors
**end function**

$x_0 \leftarrow$ Random initial values
$x_1 \leftarrow$ ALS($x_0$)
**for** $k = 1, 2, \dots$ **do**
    **if** $f(x_k) > f(x_{k-1})$ **then**
        $x_k \leftarrow x_{k-1}$
        $\beta \leftarrow 0$                                    ▷ Force ALS at this iteration
    **else**
        $\beta \leftarrow 1$
    **end if**
    $x_{k+1} \leftarrow$ ALS($x_k + \beta(x_k - x_{k-1})$)
    **if** $|f(x_{k+1}) - f(x_k)| < \delta \times f(x_k)$ **then**
        Exit loop
    **end if**
**end for**

---

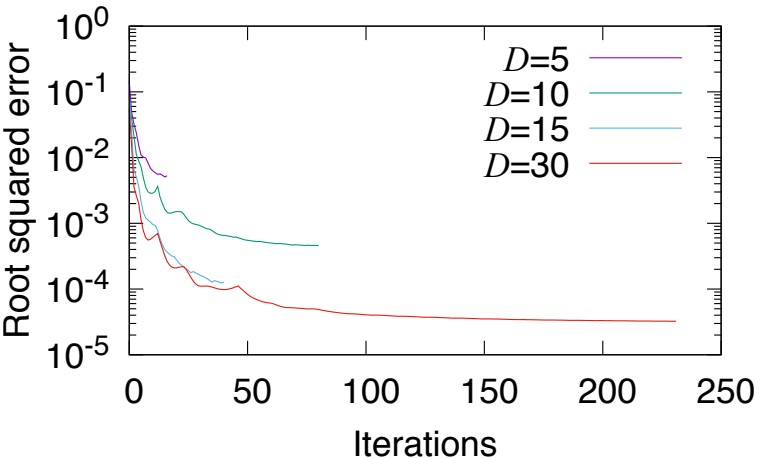

Figure 12: Convergence of root squared errors for the local susceptibility $X^{\text{loc}}$ of the single-band Hubbard model analyzed in Section 5.

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
