# Peer review of "Sparse sampling and tensor network representation of two-particle Green's functions"

_SciPost Physics, doi:SciPost Phys. 8, 012 (2020)_

## Round 1 · Referee Report · Anonymous · 2019-10-18

Strengths

1. The manuscript is well written and concisely describes the problem, the algorithmic schemes, and the examples presented in the manuscript.
2. While the manuscript deals with a specific issue with less general appeal, a solution to the problems of storage and manipulation of numerical information, using a sampling algorithm and compact representation, should be broadly interesting to a wide range of computational physicists.
3. The authors provide a bridge between two different numerical worlds, bringing tensor network representations, and associated numerical schemes developed to manipulate them, into the realm of DMFT and related methods.

Weaknesses

Minor weaknesses:
1. While the examples are sufficient to demonstrate the scheme, one really would want something more systematic, benchmarking in the single-band Hubbard model across a number of different parameters, to show the power of the technique, including changes in U and beta.
2. Computationally, one also would want a systematic analysis of the savings, particularly in storage, in applying the scheme across variations in the sampling thresholds, cutoffs, or tensor representations vs. the accuracy. Can one say anything general about limitations, at least for the simplest model?

Report

The authors present a scheme for sampling and storing information for two-particle Green's functions which alleviates two problems: (1) computational cost of sampling and (2) storage of sampled information, with sufficient accuracy. The authors extend an approach developed for the sampling of the single-particle Green's function: a sparse grid in Matsubara space. The authors also present a tensor representation for storage, which itself borrows from tensor networks used in density-matrix methods, and can borrow from the algorithms already used in that community for tasks such as optimization.

The manuscript is well written and describes the general notions of the authors' schemes for sampling and storing the information from the two-particle Green's function, which themselves are necessary for understanding the behavior of numerous response functions, from simple charge correlations to superconductivity. The authors nicely describe the extension of the SVD decomposition used in sampling the single-particle Green's function; and the authors naturally connect the representation of the Green's function to low rank tensors, providing a graphical scheme that itself should be familiar to readers from tensor network theory. The authors also provide a framework for fitting and optimizing tensors -- a cost function -- although one likely could use a number of different methods, and the manuscript itself doesn't rest on this particular point.

Using some examples, the authors demonstrate the utility of this method. While one may want a more systematic approach to the examples, really benchmarking the method and establishing some limitations/trade-offs for accuracy, computational time, and storage requirements, the examples sufficiently demonstrate what one can expect to achieve using the technique.

While the manuscript may seem specialized, the authors pin-point at least a few groups of computational scientists who may benefit by employing this method, and quite generally the ideas presented in the manuscript should be of interest to a broad slice of computational condensed matter scientists engaged in related activities. Overall, I find the manuscript to be acceptable for publication.

---

## Round 3 · Author Response

Warnings issued while processing user-supplied markup:

  • Inconsistency: Markdown and reStructuredText syntaxes are mixed. Markdown will be used.
    Add "#coerce:reST" or "#coerce:plain" as the first line of your text to force reStructuredText or no markup.
    You may also contact the helpdesk if the formatting is incorrect and you are unable to edit your text.

Dear Editor,

We thank you for your handling of our manuscript entitled "Sparse sampling and tensor network representation of two-particle Green’s function's" and the referee for his/her careful reading and feedback. The referee accepts the novelty and usefulness of the numerical methods presented in the manuscript, and recommends publication in SciPost. We made minor revisions to address the comments on minor weakness.

Best regards, Hiroshi Shinaoka Dominique Geffroy, Markus Wallerberger, Junya Otsuki, Kazuyoshi Yoshimi, Emanuel Gull, Jan Kuneš

Response to Referee:

Minor weaknesses 1

While the examples are sufficient to demonstrate the scheme, one really would want something more systematic, benchmarking in the single-band Hubbard model across a number of different parameters, to show the power of the technique, including changes in U and beta. Thank you for accepting that we presented sufficient examples to demonstrate the present scheme. Although it would be interesting to perform such systematic benchmarking, they would not fit in the current version of the manuscript of 25 pages long. Therefore, we would like to left them for future study.

Minor weaknesses 2

Computationally, one also would want a systematic analysis of the savings, particularly in storage, in applying the scheme across variations in the sampling thresholds, cutoffs, or tensor representations vs. the accuracy. Can one say anything general about limitations, at least for the simplest model? The present method is based on two independent tricks: Sparse sampling and tensor network representation. The accuracy of the sparse sampling is controlled by temperature, energy window and cutoff for singular values. These parameters also determine the size of the objects. On the other hand, the size of a tensor network can be controlled the parameter "D" (and potentially the topology of the network). It is however difficult to make a general statement on how large "D" is enough. Establishing it requires systematic benchmarking for various model as mentioned above and is beyond the scope of the present work. Instead of adding more data, we added one paragraph in Section 7 to explain how the accuracy of these techniques can be controlled.

---

## Round 3 · List of Changes

• Added one paragraph in Section 7.
  • Cited a few review articles on tensor networks and intermediate representation for better readability.

---

## Editorial Decision

published